# Learning Invariant Weights in Neural Networks

**Tycho F.A. van der Ouderaa**[1]  **Mark van der Wilk**[1]

[1]Imperial College London, UK

## Abstract

Assumptions about invariances or symmetries in data can significantly increase the predictive power of statistical models. Many commonly used machine learning models are constraint to respect certain symmetries, such as translation equivariance in convolutional neural networks, and incorporating other symmetry types is actively being studied. Yet, *learning* invariances from the data itself remains an open research problem. It has been shown that the marginal likelihood offers a principled way to learn invariances in Gaussian Processes. We propose a weight-space equivalent to this approach, by minimizing a lower bound on the marginal likelihood to learn invariances in neural networks, resulting in naturally higher performing models.

## 1 INTRODUCTION

Intuitively, invariances allow models to extrapolate, or rather 'generalise', beyond training data (see Figure 1 for an extreme example). An invariant model does not change in output when the input is changed by transformations to which it is deemed invariant. The most straightforward way to achieve this, is perhaps by enlarging the dataset with transformed examples: a process known as data augmentation. A link between invariance and data augmentation in kernel space was made by Dao et al. [2019]. We show that this invariance can equivalently be described as transformations on the weights, similar to Cohen and Welling [2016] where a neural network is constrained to respect rotational symmetry through rotated weight copies. We do what is common in Bayesian model selection and find the correct invariance using the marginal likelihood. Optimizing the marginal likelihood has proven an effective way to learn invariances in Gaussian Processes (GPs) [van der Wilk et al., 2018], but is not tractable for commonly used

neural networks. To overcome this, we propose a lower bound of the marginal likelihood capable of learning invariances in neural networks. By learning distributions on affine groups, we can select the correct invariance for a particular task, without having to perform cross-validation or even requiring a separate validation set. We succesfully learn the correct invariance on different MNIST and CIFAR-10 image classification tasks leading to better performing models.

## 2 RELATED WORK

Convolutional neural networks (CNNs) have been successful in a wide range of problems and played a key role in the success of Deep Learning [LeCun et al., 2015]. It is commonly understood that the translational symmetries that arise from effective weight-sharing in CNNs is an important driver for its outstanding performance on many tasks.

In Cohen and Welling [2016], a group-theoretical framework was proposed extending CNNs beyond translational symmetries, and demonstrated this for discrete group actions. Many studies since have proposed ways to incorporate other symmetries in neural network weights, such as continuous rotation, scale and translation, into the weights of neural networks [Worrall et al., 2017, Weiler et al., 2018, Marcos et al., 2017, Esteves et al., 2017, Weiler and Cesa, 2019, Bekkers, 2019] and recent efforts allow practical equivariance in neural networks for arbitrary symmetry groups [Finzi et al., 2021]. Nevertheless, weight symmetries are typically fixed, must be known in advance, and can not be adjusted.

Some studies have proposed invariance learning with data augmentations [Cubuk et al., 2018, Lorraine et al., 2020], but thus do not embed symmetry in weights and often require a validation loss. [Zhou et al., 2020] do learn invariances as weight-sharing, but require a meta-learning procedure with an additional validation loss. Benton et al. [2020] circumvent the need for validation data by learning a distribution of input transformations directly on the training loss. But, in doing so rely on an additional explicit regularization

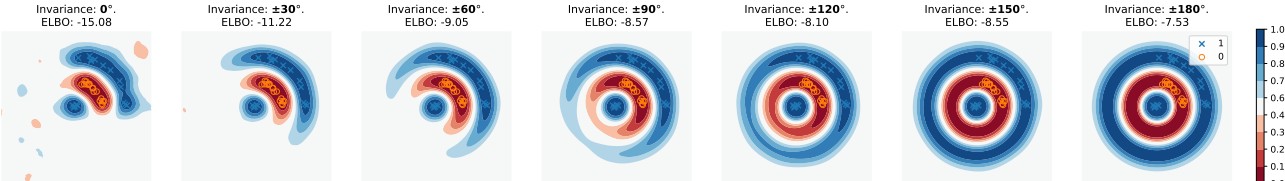

Figure 1: Illustration of extrapolating behaviour further away from toy data for models with no invariance (left), some invariance (middle) up to strict invariance (right). Model prediction plotted as contour and datapoints as ×'s and ○'s.

term that depends on how invariances are parameterised. Similar to this work, Schwöbel et al. [2020] propose to use a lower bound, but again only considers a distribution in the input space rather than on weights.

We learn invariant weights by optimizing the marginal likelihood: the common method in Bayesian statistics to perform model selection, which is parameterization independent with the aim of being generally applicable to any chosen parameterization of invariance. Interestingly, it has been shown that the marginal likelihood objective coincides with an exhaustive leave-p-out cross-validation averaged over all values of p and held-out test sets [Fong and Holmes, 2020].

Lastly, in Topological VAEs [Keller and Welling, 2021] capsules with 'rolling' feature activations show similarities to the deterministically sampled features obtained from our method, but differ in the reliance on 'temporal coherence'.

## 3  ON INVARIANT MODELLING

A model $f(\cdot)$ is deemed 'strictly invariant' its output is unaffected by a set of transformations: $f(T_g \circ \boldsymbol{x}) = f(\boldsymbol{x}), \forall g \in G, \boldsymbol{x} \in \mathcal{X}$ where each transformation $T_g$ is governed by a group action $g \in G$ forming a group $G$. We can obtain an invariant model by averaging model outputs over all transformations $T_g$. Although group theory introduces a rigid mathematical framework that is often used to describe and incorporate symmetries in statistical and machine learning models, it is restricted in the sense that the set of transformations that generate a group is always closed, by the definition of a group. To illustrate, imagine the classic MNIST image recognition problem [LeCun et al., 1998]: here invariance to rotations up to a certain angle allows for better extrapolation to tilded versions of fitted digits and thus more robust predictions and increased sample efficiency. However, invariance to full 360 degree rotations (all SO(2) group actions) may prohibit us from differentiating between a '6' and a '9'. In an effort to overcome this issue, we follow Dao et al. [2019], Raj et al. [2017], van der Wilk et al. [2018], Benton et al. [2020] and construct our invariant function $f_{\boldsymbol{\theta}}(\boldsymbol{x}; \boldsymbol{\eta})$ from a non-invariant function $g_{\boldsymbol{\theta}}(\boldsymbol{x})$ by summing over the orbit:

$$f_{\boldsymbol{\theta}}(\boldsymbol{x}; \boldsymbol{\eta}) = \int g_{\boldsymbol{\theta}}(T(\boldsymbol{x})) p_{\boldsymbol{\eta}}(T) \mathrm{d}T, \qquad (1)$$

where $p_{\boldsymbol{\eta}}(T)$ denotes a density over the group action transformations parameterised by a vector $\boldsymbol{\eta}$. Through this construction, we hope to induce a relaxed notion of invariance upon the model, sometimes referred to as 'insensitivity' [van der Wilk et al., 2018], 'soft-invariance' [Benton et al., 2020], or 'deformation stability' [Bronstein et al., 2021]. The special case in which the density $p_{\boldsymbol{\eta}}(T)$ is uniformly distributed over the orbit results in the 'Reynolds operator' from Group Theory, which averages functions and thereby induces a 'strict invariance' over the entire group.

### 3.1  INVARIANT SHALLOW NEURAL NETWORK

We construct our invariant function from a single-layer non-invariant neural network:

$$g_{\boldsymbol{\theta}}(T(\boldsymbol{x})) = \sigma\left(\boldsymbol{W}_2 \circ \phi\left(\boldsymbol{W}_1 \circ T \circ \boldsymbol{x}\right)\right), \qquad (2)$$

where $\sigma(\cdot)$ is the soft-argmax function, $\boldsymbol{x}$ is the input, and $\boldsymbol{W}_1$ and $\boldsymbol{W}_2$ are the respective first and second layer weights and biases. We omitted the bias terms for notational clarity.

In this study, we consider two flavours for our neural network $g_{\boldsymbol{\theta}}$, namely an RFF-network and ReLU-network. In the RFF-network, first layer weights $\boldsymbol{W}_1$ are initialiased as Random Fourier Reatures (RFF) [Rahimi et al., 2007] and a cosine activation function $\phi(\cdot) = \cos(\cdot)$ is used. For the ReLU-network, both first and second layer weights $\boldsymbol{W}_1$ and $\boldsymbol{W}_2$ are learned and we consider a ReLU non-linearity $\phi(x) = \max(0, x)$ for the activation function.

The RFF-network is interesting because we obtain a weight-space equivalent that is as close as possible to a GP with a radial basis function kernel (RBF), with exact correspondence in the infinite-width limit. From van der Wilk et al. [2018], we know that in this case the marginal likelihood is tight and can be used to learn invariance. The ReLU-network, on the other hand, is interesting as it more closely resembles the commonly used architectures in the Deep Learning (DL) community: basis weights are typically not fixed and the ReLU is one of the most commonly used activation functions in DL. In our experiments, we find that we can learn invariances with both the RFF-network and ReLU-network, indicating that for our purposes the bound on the marginal likelihood remains sufficiently tight for more general shallow architectures.

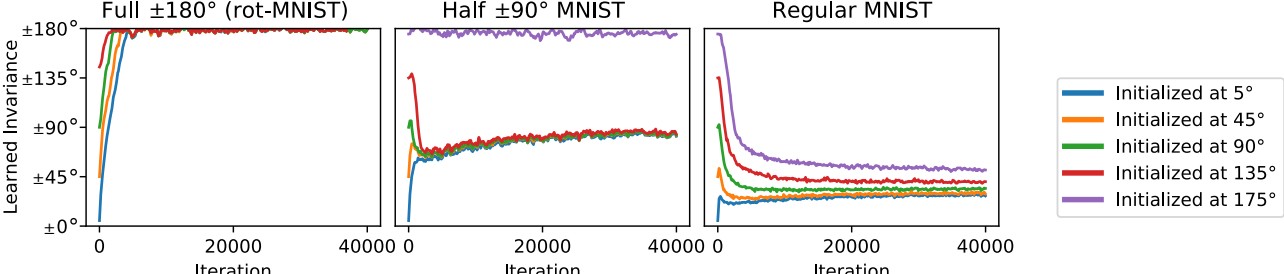

Figure 2: Predicted invariance over training iterations for models initialised with different amounts of invariance when trained on fully rotated MNIST (left), partially rotated MNIST (middle) and regular MNIST (right).

Section 3.7 will discuss how variational inference is used to learn a variational distribution $q$ over the parameters in the second layer: $\boldsymbol{\theta} = \text{vec}(\boldsymbol{W}_2)$.

## 3.2  INVARIANCE IN THE WEIGHTS

In Equations 1-2, we showed how we construct an invariant function by integrating or summing over transformed input samples $T(\boldsymbol{x})$. Yet, instead of explicitly performing these transformations on the input, we can obtain a mathematically equivalent invariant function by considering transformations on the weights. Note that the inner term of our neural network definition in Equation 2, we have that $(\boldsymbol{W}_1 \circ T) \circ \boldsymbol{x} = \boldsymbol{W}_1 \circ (T \circ \boldsymbol{x})$ are equal, by associativity of matrix transformations. In other words, first applying transformation $T$ on the weights, similar to the typical construction of equivariant layers, is equivalent to first applying it to the input, which could be interpret as built-in data augmentation. In practice, however, differences between the two could still arise if applying $T$ requires approximations (e.g. interpolation between discrete pixels). In our experiments we will consider transforming the weights, thus demonstrating that invariance can be 'built into' the model.

**Coordinate data and imaging data**   We consider simple affine transformations, which can be represented as $T \in \mathbb{R}^{3 \times 3}$ matrices. For 2d vector data, applying the transformations amounts to regular matrix multiplications, which only requires appending a single 1-entry to the data vectors. For 2d images, where data points $\boldsymbol{x} \in \mathbb{R}^{WH}$ correspond to $W \times H$ pixel grids, applying $T$ in image space requires interpolation. Here, we could use bilinear interpolation, which can also be written in matrix formula form. We use the grid sample operation (as used in [Jaderberg et al., 2015]) which acts on the weight matrix values $\boldsymbol{W}_1$ and outputs an equally shaped matrix. The operation treats the $HW$-dimensional row vectors of $\boldsymbol{W}_1$ as a grid of $H \times W$ points where the coordinates of the point are transformed according to the affine transformation matrix $T$. The resulting values are obtained by interpolating the values of transformed pixels at the original grid coordinates using bilinear interpolation.

## 3.3  AFFINE LIE GROUP REPARAMETERIZATION

The transformations that are applied on the weights and will define the invariances of the network are sampled from a probability distribution. To allow learnable invariances, we define a learnable probability distribution over the transformations $p_{\boldsymbol{\eta}}(T)$ parameterised by $\boldsymbol{\eta}$. We will refer to $\boldsymbol{\eta}$ as the 'invariance parameters', as they parameterise to which transformations to which our network becomes invariant. To learn this distribution with back-propagation, we must make sure that samples taken from the distribution are differentiable with respect to the invariance parameter $\boldsymbol{\eta}$. For affine transformed weights, we consider a procedure similar to what Benton et al. [2020] used to augment inputs, utilising the re-parameterization trick [Kingma and Welling, 2013] to remain differentiable. The distribution defines independent Gaussian probabilities over infinitesimal generators around their origin. sampling noise from a k-cubed uniform distribution $\boldsymbol{\epsilon} \sim U[-1, 1]^k$. With $k$=6 generator matrices $\boldsymbol{G}_1, \cdots, \boldsymbol{G}_6$ and learnable parameters $\boldsymbol{\eta} = [\eta_1, \cdots, \eta_6]^T$ we can separately parameterise translation in x, translations in y, rotations, scaling in x, scaling in y, and shearing (see Appendix D). A sample $T \sim p_{\boldsymbol{\eta}}(T)$ can be obtained by transforming noise $\boldsymbol{\epsilon}$:

$$T = \exp\left(\sum_i \epsilon_i \eta_i \boldsymbol{G}_i\right), \qquad \boldsymbol{\epsilon} \sim U[-1, 1]^k \qquad (3)$$

with matrix exponential $\exp(M) = \sum_{n=0}^{\infty} \frac{1}{n!} M^n$. A distribution over the subgroup of 2d rotations SO(2) can be achieved by only learning the parameter for rotational invariance $\eta_{\text{rot}} = \eta_3$ and fixing $\eta_i = 0$ for all $i \neq 3$. Then,

$$T^{(\text{rot})} = \begin{bmatrix} \cos(\epsilon_3 \eta_{\text{rot}}) & -\sin(\epsilon_3 \eta_{\text{rot}}) & 0 \\ \sin(\epsilon_3 \eta_{\text{rot}}) & \cos(\epsilon_3 \eta_{\text{rot}}) & 0 \\ 0 & 0 & 1 \end{bmatrix} \qquad (4)$$

By learning $\eta_{\text{rot}}$, we can effectively interpolate between no invariance at $\eta_{\text{rot}} = 0$ to full rotational invariance at $\eta_{\text{rot}} \equiv \pi$.

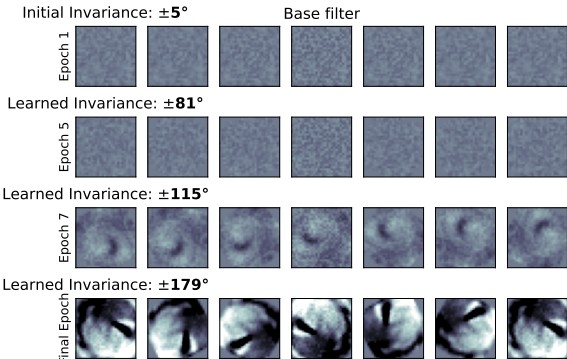

(a) Feature bank #1 over training iterations.

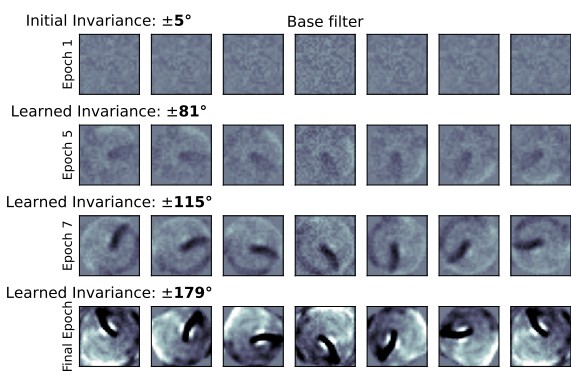

(b) Feature bank #2 over training iterations.

Figure 3: Illustration of converging filter banks of two features. Features are initialised randomly with almost no invariance and converge to particular filters with practically full ($\pm 179$) rotational invariance after training on the fully-rotated MNIST.

Similarly, we can define a distribution over the subgroup of 2d translations $\mathbb{T}(2)$ by fixing $\eta_i = 0$ for all $i > 2$ and learning the translational invariance parameters $\eta_1$ and $\eta_2$:

$$T^{(\text{trans})} = \begin{bmatrix} 1 & 0 & \epsilon_1 \eta_1 \\ 0 & 1 & \epsilon_2 \eta_2 \\ 0 & 0 & 1 \end{bmatrix} \qquad (5)$$

We include full derivations including scaling in Appendix E. In general, $\boldsymbol{\eta} = \mathbf{0}$ corresponds to no invariance and increasing individual elements of $\boldsymbol{\eta}$ also increases insensitivity to corresponding transformations towards full invariance.

### 3.4 STOCHASTIC OR DETERMINISTIC SAMPLING

To estimate $f_{\boldsymbol{\theta}}(x; \boldsymbol{\eta})$ from Equation 1, we approximate the integral with a Monte Carlo (MC) estimate:

$$\hat{f}_{\boldsymbol{\theta}}(\boldsymbol{x}; \boldsymbol{\eta}) = \frac{1}{S} \sum_{i=1}^{S} g_{\boldsymbol{\theta}}(T_i(\boldsymbol{x})) \qquad (6)$$

where $S$ transformations are stochastically sampled from the distribution $T_i \sim p_{\boldsymbol{\eta}}(T)$. Samples can be differentiated with respect to invariance parameter $\boldsymbol{\eta}$ using the 're-parameterization trick' (see Section 3.3). We know that MC is an unbiased estimator, and thus

$$f_{\boldsymbol{\theta}}(\boldsymbol{x}; \boldsymbol{\eta}) = \mathbb{E}_T \left[ \hat{f}_{\boldsymbol{\theta}}(\boldsymbol{x}; \boldsymbol{\eta}) \right] \qquad (7)$$

with $\mathbb{E}_T := \mathbb{E}_{\prod_{i=1}^{S} p_{\boldsymbol{\eta}}(T_i)}$. Unlike stochastic MC sampling, we can obtain a deterministic surrogate of the procedure by replacing the stochastic samples from the noise source $U[-1, 1]^k$ with linearly spaced points along its $k$-cubed domain. This procedure is similar to quadrature in classical

numerical integration, or from a programming perspective, as applying the re-parameterization trick on a fixed 'linspace' instead of uniform noise. A visualization of a discretely sampled filter bank of a model learning rotational invariance over training iterations is shown in Figure 3. By ensuring sufficient and equally spaced samples, deterministic sampling can be used to ensure reliable and robust inference at test time. Similar to the stochastic sampling, this deterministic procedure is also differentiable and can thus be used during training. We find, however, that deterministic sampling is only suitable when the number of invariances $\dim(\boldsymbol{\eta})$ is very small (see Section 3.5). Nevertheless, deterministic sampling can be theoretically interesting and allow our model to be interpret as a generalization of other architectures. For instance, a single convolutional layer where the kernel is discretely and deterministically convolved over an image followed by spatial pooling, can be interpret as an instance of our invariant MLP with a specific affine invariance transformation in which weights are 'zoomed-in' and deterministically sampled and reapplied over the image plane.

### 3.5 PRACTICAL TRANSFORMATION SAMPLING

If $\boldsymbol{\eta}$ comprises multiple elements, the sampling suffers from the curse of dimensionality as the number of required samples grows exponentially with larger $K$. To illustrate, a sparse 3 quadrature points in $K = 6$ dimensions would already require $3^6 = 729$ samples with deterministic sampling. In general, we found that stochastic MC sampling resulted in the most stable training behaviour and therefore used this when training the models in the experimental section, except for Figure 3 where deterministic sampling was used for both training and visualization of rotationally invariant filter bank.

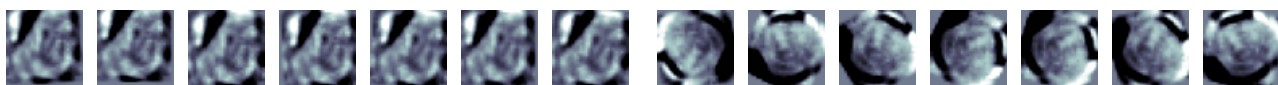

(a) Sampled filters of affine model trained on regular mnist.    (b) Sampled filters of affine model trained on rotated mnist.

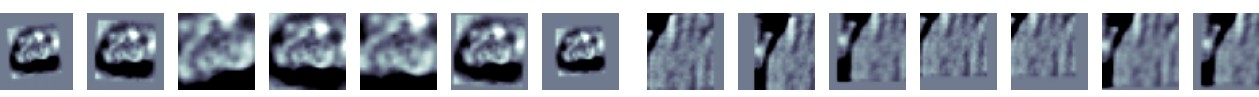

(c) Sampled filters of affine model trained on scaled mnist.    (d) Sampled filters of affine model trained on translated mnist.

Figure 4: Stochastic samples of learned filter banks of a model capable of learning affine invariances. The same model learns features that are insensitive to different kinds of transformations dependent on the data it was trained on.

### 3.6 LOWER BOUNDING THE MARGINAL LIKELIHOOD

We have a (typically large) vector $\boldsymbol{\theta}$ containing the model parameters and a (typically small) vector for the invariance parameters $\boldsymbol{\eta}$. The approach we take in this paper is to perform Bayesian Model Selection and integrate out $\boldsymbol{\theta}$ but find a point-estimate over $\boldsymbol{\eta}$:

$$\hat{\boldsymbol{\eta}} = \arg\max_{\boldsymbol{\eta}} p(\mathcal{D}|\boldsymbol{\eta}) = \arg\max_{\boldsymbol{\eta}} \left[ \int p(\mathcal{D}|\boldsymbol{\theta}) p(\boldsymbol{\theta}|\boldsymbol{\eta}) \mathrm{d}\boldsymbol{\theta} \right] \tag{8}$$

where $p(\mathcal{D}|\boldsymbol{\eta})$ is the marginal likelihood [Murphy, 2012] or model evidence, sometimes called empirical Bayes or type-II ML. The technique has been shown effective in GPs to learn hyper-parameters Williams and Rasmussen [2006] and invariances van der Wilk et al. [2018], but is typically intractable for neural networks. We derive a lower bound that allows for optimization of the marginal likelihood in neural networks using stochastic variational inference:

$$\log p(\mathcal{D}) \geq \mathbb{E}_{\boldsymbol{\theta}} \left[ \log p(\mathcal{D}|\boldsymbol{\theta}) \right] - \mathrm{KL}(q(\boldsymbol{\theta}|\boldsymbol{\mu}, \boldsymbol{\Sigma}) || p(\boldsymbol{\theta}))$$
$$= \mathbb{E}_{\boldsymbol{\theta}} \left[ \log p(\boldsymbol{y}|f_{\boldsymbol{\theta}}(\boldsymbol{x}; \boldsymbol{\eta})) \right] - \mathrm{KL}(q(\boldsymbol{\theta}|\boldsymbol{\mu}, \boldsymbol{\Sigma}) || p(\boldsymbol{\theta}))$$
$$= \mathbb{E}_{\boldsymbol{\theta}} \left[ \log p \left( \boldsymbol{y} \middle| \mathbb{E}_T \left[ \hat{f}_{\boldsymbol{\theta}}(\boldsymbol{x}; \boldsymbol{\eta}) \right] \right) \right] - \mathrm{KL}(q(\boldsymbol{\theta}|\boldsymbol{\mu}, \boldsymbol{\Sigma}) || p(\boldsymbol{\theta}))$$
$$\geq \mathbb{E}_{\boldsymbol{\theta}} \left[ \mathbb{E}_T \left[ \log p(\boldsymbol{y}|\hat{f}_{\boldsymbol{\theta}}(\boldsymbol{x}; \boldsymbol{\eta})) \right] \right] - \mathrm{KL}(q(\boldsymbol{\theta}|\boldsymbol{\mu}, \boldsymbol{\Sigma}) || p(\boldsymbol{\theta})) \tag{9}$$

with expectations $\mathbb{E}_{\boldsymbol{\theta}} := \mathbb{E}_{q(\boldsymbol{\theta})}$ and $\mathbb{E}_T := \mathbb{E}_{\prod_{i=1}^{S} p_{\boldsymbol{\eta}}(T_i)}$. We begin Eq 9 with the standard evidence lower bound (ELBO) derived from variational inference. In the second and third line, we expand the likelihood and plug-in Equation 7. In the last line, we use Jensen's inequality together with the fact that our log-likelihood is a convex function. The resulting lower bound comprises an expected log-likelihood term that can be estimated by taking the average cross-entropy on mini-batches of data (see Section 3.7) and a KL-divergence between two multivariate Gaussians which can efficiently be computed in closed-form. Note, we integrate out model parameter vector $\boldsymbol{\theta}$, which is part of the KL-term, whereas the vector parameterizing the invariances $\boldsymbol{\eta}$ is only part of the first term. We optimise the derived lower bound w.r.t. both $\boldsymbol{\eta}$ and $\boldsymbol{\theta}$ every iteration with stochastic gradient descent.

### 3.7 VARIATIONAL INFERENCE

To summarise, we propose to learn invariances using stochastic variational inference [Hoffman et al., 2013] and derived a lower bound of the marginal likelihood, or *evidence lower bound* (ELBO) that can be optimised using a gradient descent methods, such as Adam [Kingma and Ba, 2014]. Variational inference minimises the KL-divergence between an variational posterior and the true posterior on our free model parameters $p(\boldsymbol{\theta}|\mathcal{D})$, where $\boldsymbol{\theta} = \mathrm{vec}(\boldsymbol{W}_2)$. For the approximate posterior, we choose a multivariate Gaussian distribution $q(\boldsymbol{\theta}|\boldsymbol{\mu}, \boldsymbol{\Sigma}) := \mathcal{N}(\boldsymbol{\theta}|\boldsymbol{\mu}, \boldsymbol{\Sigma})$ parameterised by variational parameters $\boldsymbol{\mu}$ and block-diagonal covariance $\boldsymbol{\Sigma}$ with a separate block for each output class. The covariance is parameterised as a Cholesky decomposition $\boldsymbol{\Sigma} = \boldsymbol{L}^T \boldsymbol{L}$, which is a common trick to maintain computational stability to ensure a positive semi-definite $\boldsymbol{\Sigma}$ and does not influence the model. We obtain a differentiable Monte Carlo estimate of $q(\boldsymbol{\theta})$ by sampling $L$ times from the variational distribution, using the reparameterization trick [Kingma and Welling, 2013], and maximise the ELBO:

$$\mathcal{L} = \mathbb{E}_{\boldsymbol{\theta}} \left[ \mathbb{E}_T \left[ \log p(\boldsymbol{y}|\hat{f}_{\boldsymbol{\theta}}(\boldsymbol{x}; \boldsymbol{\eta})) \right] \right] - \mathrm{KL}(q(\boldsymbol{\theta}|\boldsymbol{\mu}, \boldsymbol{\Sigma}) || p(\boldsymbol{\theta}))$$
$$\approx \frac{1}{L} \sum_{l=1}^{L} \Big[ \underbrace{\log p(\boldsymbol{y}|\frac{1}{S} \sum_{i=1}^{S} g_{\boldsymbol{\theta}_l}(T_i(\boldsymbol{x})))}_{\text{Cross-entropy}} \Big] - \underbrace{\mathrm{KL}(q(\boldsymbol{\theta}) || p(\boldsymbol{\theta}))}_{\text{Closed-form KL}} \tag{10}$$

where we can choose $L\{=\}1$ given a sufficiently large batch size. We obtain a Stochastic Gradient Variational Bayes (SGVB) estimate of the lower bound $\frac{N}{M} \sum_{i=1}^{M} \tilde{\mathcal{L}}(\boldsymbol{\theta}, \{\boldsymbol{x}_i\}, \{y_i\})$ [Kingma and Welling, 2013] to allow efficient training on mini batches of data. Full derivations can be found in Appendix A.

| | Test Accuracy | | | ELBO | | |
|---|---|---|---|---|---|---|
| | Fully rotated | Partially rotated | Regular | Fully rotated | Partially rotated | Regular |
| Model | MNIST | MNIST | MNIST | MNIST | MNIST | MNIST |
| MLP + fixed 5° rotation | 79.29 | 86.71 | **96.00** | -1.07 | -0.80 | -0.36 |
| MLP + fixed 45° rotation | 87.35 | 91.13 | 95.93 | -0.63 | -0.49 | **-0.26** |
| MLP + fixed 90° rotation | 90.33 | **91.69** | 94.69 | -0.52 | **-0.44** | -0.30 |
| MLP + fixed 135° rotation | 91.19 | 91.04 | 92.13 | -0.45 | -0.45 | -0.36 |
| MLP + fixed 175° rotation | **91.57** | 90.47 | 90.97 | **-0.43** | -0.47 | -0.45 |
| MLP + learned rotation | **91.72** | **92.34** | **96.40** | **-0.43** | **-0.42** | **-0.26** |

Table 1: Test Accuracy and ELBO scores on MNIST using RFF neural network[1]. For each dataset, we observe that the correct level of invariance for that dataset corresponds with highest ELBO and also correlates with best test accuracy. In addition, we find that automatically learned invariance converges to ELBO and test accuracies similar or beyond the found optimal values from the models with fixed invariance.

# 4 EXPERIMENTS AND RESULTS

We implemented our method in PyTorch [Paszke et al., 2017] and show results on a toy problem with different degrees of rotational invariance in Figure 1 with 1024 RFF features, $\sigma = 5$, and $T$ applied on the weights.

The following sections will describe experiments on different MNIST and CIFAR-10 image classification tasks where $T$ is applied on the weights by using the bilinear grid resampling as described in Section 3.2 and Jaderberg et al. [2015] in combination with small 0.1 sigma Gaussian blur to bandlimit high frequencies. We used Adam [Kingma and Ba, 2014] for optimization in combination with a learning rate of 0.001 ($\beta_1 = 0.9, \beta_2 = 0.999$) cosine annealed [Loshchilov and Hutter, 2016] to zero. Parameters were initialised as $\boldsymbol{\mu}_c = \boldsymbol{0}$, $\boldsymbol{L}_c = \boldsymbol{I}$ for all classes $c$, $\sigma = 0.3$, and $\alpha = 1.0$. We use $S = 32$ samples from $p_{\boldsymbol{\eta}}(T)$, $L = 1$ and a batch size of 128.

## 4.1 ON THE NECESSITY OF A BAYESIAN APPROACH

To investigate to what extent the variational inference is required to learn invariances, we compare our approach with regular maximum likelihood using Adam. We train one model that uses our objective (Variational Inference), and another model where we replaced the variational distribution $q(\boldsymbol{\theta})$ with a point-estimate and omitted the KL-term to get a regular cross-entropy loss. Interestingly, when trained on fully-rotated MNIST in Figure 5, we find that the model trained with cross-entropy was completely incapable of learning the correct invariance, whereas our VI-based approach does learn the invariance. We hypothesise that maximum likelihood alone is not enough to learn invariance, as invariance is a constraint on the weights and thus does not help to fit the data better, whereas marginal likelihood also favours simpler models. This result substantiates the use of marginal likelihood (or a lower bound thereof) for hyper-parameter selection for neural networks, and invari-

ance learning in particular. More broadly speaking, it proves a convincing case for probabilistic machine learning models, such as Bayesian neural networks, beyond their oft-cited use for uncertainty estimation.

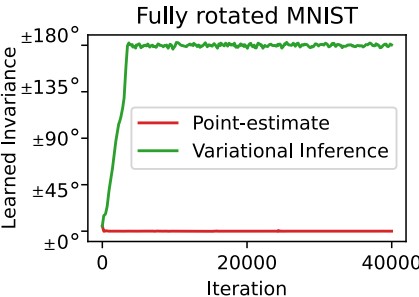

Figure 5: Predicted invariance over training iterations with non-Bayesian point-estimate optimised with cross-entropy and approximate Bayesian inference. A regular point estimate can not learn invariances, whereas our VI-based approach does learn the invariance.

## 4.2 IDENTIFYING INVARIANCE WITH ELBO

To evaluate whether the ELBO is capable of identifying the apt level of invariance, we consider models with different fixed values of rotational invariance $\eta_{\text{rot}}$ and one model where $\eta_{\text{rot}}$ is learned. We then evaluate the models on three different versions of MNIST on which we artificially imposed different amounts of rotational invariance by randomly transforming the dataset beforehand. In 'fully rotated MNIST', we rotate every image with a random uniformly sampled angle in range $[-180°, 180°]$. In 'partially rotated MNIST' images are rotated with a random angle within $[-90°, 90°]$. Lastly, we also consider the 'regular MNIST' dataset without any alterations.

---

[1] As explained in Section 3.1, we use the RFF neural network to ensure a tight lower bound and for comparison purposes. Higher accuracies on MNIST and CIFAR-10 were achieved with a ReLU neural network as reported in Table 2 and Table 3.

From Table 1, we observe that for each dataset the model with the best ELBO corresponds to the model with the right amount of invariance, also correlating with best test accuracy. This finding indicates that the ELBO can correctly identify the required level of invariance, and confirms that choosing the right invariance leads to better generalization on the test set. On regular MNIST, we observe that a small amount of invariance yields better ELBO than no invariance. This could be explained by some intrinsic rotational variation within the dataset. Furthermore, we find that the ELBO of the model with learned invariance $\eta_{\mathrm{rot}}$ corresponds to the optimal ELBO in the set of models with fixed invariance. Therefore, we find that in this case, we can use the ELBO to learn invariances in a differentiable manner. Additional results can be found in Appendix C.

## 4.3 RECOVERING INVARIANCE FROM INITIAL CONDITIONS

To investigate robustness to different initial conditions, we repeat the experiment where we learn invariance parameters $\eta$ during during training on fully-rotated, partially rotated and regular MNIST data but with different initial values, corresponding to rotational invariance of $[\pm 5°, \pm 45°, \pm 90°, \pm 135°, \pm 175°]$ degrees. Results of this experiments for the RFF neural network are shown in Figure 2, and a similar figure for the ReLU neural network is attached in Appendix C.2. For most initial conditions, we observe that we can succesfully learn and recover the 'correct' amount of invariance for each dataset. One exception being initial $175°$ degrees on partially rotated dataset, which suggests that training with low initial invariance could be advantageous in practice, for this method. Nevertheless, we conclude that our model can recover invariance relatively robustly independent of initial conditions.

## 4.4 LEARNING INVARIANCE IN RELU NETWORK

So far, we have only considered the set-up where we learn the output layer $W_2$ and keep the first layer $W_1$ initialised as fixed RFF-features in combination with a cosine $\cos(\cdot)$ activation function. We chose this fixed basis function model to ensure a sufficiently tight bound on marginal likelihood where the only source of looseness is the non-Gaussian likelihood. Now, we will let loose of these constraints and consider a general single hidden layer neural network with ReLU non-linearity $\phi(x) = \max(x, 0)$ with Xavier [Kumar, 2017] initialised weights and 1024 hidden units, where we learn both the input layer $W_1$ the output layer $W_2$. We optimise the model using the same variational inference procedure.

We find that we are still able to learn invariances in the setting where parameters of both input and output layer are

learned (full comparison in Appendix C). In Figure 3, we plot an illustration of a feature bank (row vector in $W_1$ with 7 samples equally spaced between $-\eta_{\mathrm{rot}}$ and $\eta_{\mathrm{rot}}$ and plotted over training iterations). The top of the figure shows the randomly initialised features without any rotational invariance at the beginning of training. After training on a fully rotated MNIST, the features converge to a particular filter with practically full $\pm 179°$ rotational invariance, as shown on the bottom of the same figure.

## 4.5 OTHER TRANSFORMATIONS

To explore invariance to transformations other than rotation, we allow for different kinds of affine invariance transformations, namely rotation, translation, scale and full affine transformations (see Section 3.3). Again, we use the ReLU-network where both layers are learned.

In Table 2, we evaluate and compare models that can learn affine invariances with two non-invariant baselines, namely a regular Gaussian Process regression with RBF kernel baseline (SGPR) and a regular shallow neural network baseline (MLP). We use SGPR as a reference, because we know the training procedure is reliable and to ensure enough capacity is given to the single layer MLP. We separately trained the models on fully-rotated, translated, scaled and original versions of MNIST (see Appendix D for details). We find that models with learned invariances (bottom four rows) outperform the model with no invariance (top two rows) in all cases. As expected, a translationally invariant model performs better on a dataset that contains randomly translated examples, and similarly, the rotationally and scale invariant models perform best on the respective rotated and scaled versions of MNIST. In line with our expectations, the model capable of learning affine invariances performs best overall. Moreover, by inspecting the learned coefficients of $\eta$ after training we verified that the learned transformations correspond to the dataset the it was trained on. This can also be observed in Figure 4 by inspecting the resulting learned filter banks samples after training on different datasets.

|  | Test Accuracy | | | |
| Model | Fully rotated MNIST | Translated MNIST | Scaled MNIST | Regular MNIST |
| --- | --- | --- | --- | --- |
| SGPR | 91.19 | 89.22 | 72.10 | 97.52 |
| MLP | 90.35 | 89.34 | 96.61 | 98.10 |
| MLP + Rotation (**ours**) | **98.05** | 94.08 | 97.62 | 98.64 |
| MLP + Translation (**ours**) | 93.59 | **97.87** | 97.98 | 98.76 |
| MLP + Scale (**ours**) | 93.80 | 94.30 | **98.06** | 98.35 |
| MLP + Affine (**ours**) | 98.14 | 97.66 | 98.31 | **98.93** |

Table 2: Test Accuracy scores for learned invariance using different transformations in a shallow ReLU neural network on the MNIST dataset.

We repeated the same experiment on the CIFAR-10 dataset Krizhevsky et al. [2009] and trained on fully-rotated, translated, scaled version and the original version of the

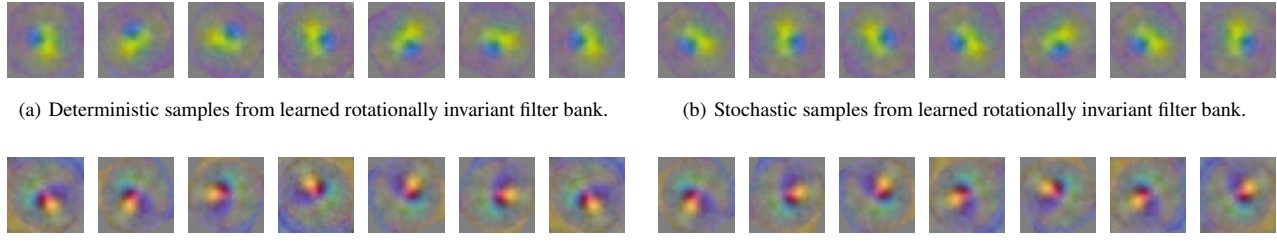

(a) Deterministic samples from learned rotationally invariant filter bank.

(b) Stochastic samples from learned rotationally invariant filter bank.

(c) Deterministic samples from learned rotationally invariant filter bank.

(d) Stochastic samples from learned rotationally invariant filter bank.

Figure 6: Visualization of samples from learned filter banks using discrete sampling learned on different versions of CIFAR-10. The invariant transformations are learned starting from no invariance dependent on the data it was trained on.

CIFAR-10 dataset and plot test accuracies in Table 3. We consistently find that the best performing models are those that are parameterised such that it can learn the invariance that corresponds the dataset, typically resulting in several percentage points of improved accuracy compared to the MLP baseline. Furthermore, if we parameterise the MLP with the more general affine invariance, capable of expressing rotation, translation and scale invariances, we always achieve similar or improved results compared to model from the models parameterised with a a single invariance. Similar to the MNIST experiments, we find that the MLP with general affine invariances can select the correct invariance based on the used training data. Here we also verified this by inspecting the $\boldsymbol{\theta}$, and found that the learned invariance always matches the invariances that we expect for the corresponding dataset. For example, the model capable of learning affine invariances correctly learned to be rotationally invariant ($\eta_3 \approx \pi$ and $\eta_i \approx 0$ for $i \neq 3$) after training on the fully-rotated CIFAR-10 dataset.

| | Test Accuracy | | | |
|---|---|---|---|---|
| Model | Fully rotated CIFAR-10 | Translated CIFAR-10 | Scaled CIFAR-10 | Regular CIFAR-10 |
| MLP | 41.24 | 40.75 | 46.56 | 54.49 |
| MLP + Rotation (**ours**) | **46.04** | 40.71 | 46.77 | 54.72 |
| MLP + Translation (**ours**) | 40.99 | **45.20** | 47.44 | **55.79** |
| MLP + Scale (**ours**) | 40.92 | 41.22 | **49.28** | 54.72 |
| MLP + Affine (**ours**) | **46.12** | 45.77 | 48.81 | **55.44** |

Table 3: Test Accuracy scores for learned invariance using different transformations in a shallow ReLU neural network on the CIFAR-10 dataset.

## 5 DISCUSSION AND CONCLUSION

In this paper, we propose a single training procedure capable that can *learn* invariant weights in neural networks automatically from data. We follow what is common in Bayesian statistics and optimise the marginal likelihood to perform Bayesian model selection: a method that has been proven capable to learn invariances in GPs [van der Wilk et al., 2018]. We propose a lower bound to allow optimization of the *marginal likelihood* in shallow neural networks.

On MNIST and CIFAR-10 image classification tasks, we demonstrate that we can automatically learn weights that are invariant to correct correct affine transformations, solely using training data. Furthermore, we show that this leads to better generalization and higher predictive test accuracies.

The marginal likelihood is a general model selection method and is parameterization independent. Therefore, we can expect it to work on other invariances and other model architectures. In this work, we focussed on affine transformations, but it would be interesting to consider more complex parameterizable transformations over the image space, such as diffeomorphic vector fields [Schwöbel et al., 2020]. We showed that we can learn invariance by sampling a learned compactly supported continuous probability distribution over group actions in common Lie groups. Allowing discrete groups would either require differentiating through a discrete probability distribution, for instance utilizing the Gumbel-Softmax trick [Jang et al., 2016]), or, by treating the discrete group as a subgroup of some Lie group and learn to approximately distribute all continuous density $p_{\boldsymbol{\eta}}(T)$ to the group actions of the subgroup. Furthermore, it would be interesting to consider more flexible and complex probability densities over group actions, such as a mixture distributions or normalizing flows [Rezende and Mohamed, 2015, Tabak and Turner, 2013], capable of expressing multiple modes as in [Falorsi et al., 2019]. We found that we could succesfully learn invariance using marginal likelihood, also referred to as Empirical Bayes or Type-II ML, which is not possible with regular maximum likelihood (Type-I ML). To do so, we relied on $\boldsymbol{\eta}$ being small and learning higher dimensional invariances might therefore require more sophisticated methods or additional priors on $\boldsymbol{\eta}$. Lastly, this work focuses on single layer neural networks, and we will consider deeper architectures in future work. For deeper models, we should ask the question whether the bound on the marginal likelihood will stay sufficiently tight [Dutordoir et al., 2021, Ober and Aitchison, 2020, Immer et al., 2021].

To conclude, we hope our findings inspire other works to allow neural networks that automatically learn symmetries from data.

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
