# OpenReview forum: "Learning Invariant Weights in Neural Networks"
_auai.org/UAI/2022/Conference — UAI 2022 Oral_

### Official Review · Reviewer_Cq9v · 2022-03-19

**Q2(1) Originality/Novelty:** 2
**Q2(2) Significance/Impact:** 2
**Q2(3) Correctness/Technical Quality:** 3
**Q2(6) Clarity Of Writing:** 2
**Q6 Overall Score:** 5
**Q8 Confidence In Your Score:** 3

**Q1 Summary And Contributions:**

This paper studies the problem of learning invariances in neural networks. Here invariance is with respect to transformations to the input, i.e. the paper gives the example of rotated MNIST. The paper proposes a method to learn a distribution over the transformations, and takes the prediction averaged over this distribution as the output of the model. Experimental results for rotated MNIST show that this method can be used to obtain comparable likelihood/accuracy as encoding the invariance.


**Q2 Assessment Of The Paper:**

More detailed information regarding each of these aspects is given below:

**Q2(4) Quality Of Experiments (Optional):**

3: Good: The experimental evaluation is adequate, and the results convincingly support the main claims.

**Q2(5) Reproducibility:**

2: Fair: Key resources (e.g., proofs, code, data) are unavailable but key details (e.g., proof sketches, experimental setup) are sufficiently well-described for an expert to confidently reproduce the main results.

**Q3 Main Strengths:**

The premise of the motivating question is interesting: if we can automatically learn invariances in the data, that could be very helpful in terms of developing more robust and generalizable algorithms. This paper takes a first step towards tackling this question and comes up with a method with some empirical support.

**Q4 Main Weakness:**

The main weakness of this work is with regards to the scalability and potential impact of the proposed method. It is unclear whether the proposed method is very scalable. The method relies on learning a distribution over potential transformations to the input and taking the average output of the model over this distribution. This can work if the space of transformations is limited to very common ones, such as rotation, translation, etc., but it is unclear if this will scale as the space of transformations grows. Furthermore, only 2 layer neural nets are considered in this work.

**Q5 Detailed Comments To The Authors:**

- It is concerning that the method doesn't work for point estimates and requires a Bayesian approach, as in terms of expressivity the point estimate model should seemingly be sufficient. Are there any more specific reasons for the failure, i.e., poor optimization, etc.?
- The terminology "invariant weights" is a bit misleading since the invariance doesn't really come from the weights, but instead an architectural design choice. (Because the proposed model averages over a set of fixed transformations to produce the invariance, and the learnt part is simply the distribution over the transformations).
- The current work applies to the setting where the transformations for the invariance are already known in advance, and what remains to be chosen is the degree of the invariance. Is there any chance of extending this work to learning the class of transformations as well?
- What is the motivation of studying 2 layer NN's with fixed first layer and ReLU networks separately?

**Q7 Justification For Your Score:**

I think this paper is borderline because the scalability of this method is unclear -- it seems to work in settings where there is a specific set of transformations in mind, but it is not clear how well the method will do for larger space of transformations and training larger models. In the current limited scope of invariances considered, it is not clear how useful it is to learn invariances, because for datasets considered, the optimal invariance is known and could be encoded in advance.

**Q9 Complying With Reviewing Instructions:**

1: Yes.

---

### Official Review · Reviewer_pdYu · 2022-04-11

**Q2(1) Originality/Novelty:** 3
**Q2(2) Significance/Impact:** 2
**Q2(3) Correctness/Technical Quality:** 3
**Q2(6) Clarity Of Writing:** 3
**Q6 Overall Score:** 7
**Q8 Confidence In Your Score:** 4

**Q1 Summary And Contributions:**

Introducing simmetries into a model has proven to be a powerful approach in Machine Learning, for example, convolutional neural networks work well for images due to their translation invariance. This work proposes to learn these invariances in a data-driven fashion and, instead of modifying the input data, learn a weight-space equivalent.

The approach is validated in several image datasets, showing how the model is able to learn the proper invariance depending on the dataset.

**Q2 Assessment Of The Paper:**

More detailed information regarding each of these aspects is given below:

**Q2(4) Quality Of Experiments (Optional):**

3: Good: The experimental evaluation is adequate, and the results convincingly support the main claims.

**Q2(5) Reproducibility:**

2: Fair: Key resources (e.g., proofs, code, data) are unavailable but key details (e.g., proof sketches, experimental setup) are sufficiently well-described for an expert to confidently reproduce the main results.

**Q3 Main Strengths:**

- The idea presented is really appealing and has a lot of potential, learning invariances depending on the data would open a lot of possiblities.
- The proposed approach is clear and simple to follow.
- The experiments are sensible and well-designed to show the points the work want to prove.
- Plenty of figures help understand the results, and appendix E helps a lot understanding finer details.

**Q4 Main Weakness:**

- 4.1 While I understand the motivation (CNN as an example), I find it a bit weak in this paper. Why do we need to learn invariances on something more than the input/first layer? What are the benefits?
- 4.2 There are no comparisons with existing approaches (e.g. with Schwöbel et al. [2020]) and I miss some more baselines (e.g., a small CNN) and additional not-so-favorable experiments to test the limits of the method (e.g., what happens if the data contains more than one simmetry?)
- 4.3 Presentation could be improved (several typos, wrong citation format, phrases excesively long without commas, etc.).
- 4.4 I could not find some important details (see below).


**Q5 Detailed Comments To The Authors:**

- 5.1 I have been looking throught the entire paper, but I cannot fully grasp the way the proposed shallow network works. What I mean is that, a flattened MNIST image has 784 dimensions, therefore the weight matrix W will have 784 columns. However, the matrices T presented through the entire paper are of size 3x3. Could you clarify how are you exactly using T within the network?
- 5.2 I think this is a typo, but why don't you use the last expression in Eq. 9 at the end of the same page?

**Q7 Justification For Your Score:**

The idea of the paper is quite interesting. However, the motivation to the solution is rather weak, and I am missing different baselines and experiments that would enrich the work.

Pros outweigh cons in my opinion, and given how lightweight experiments are, the authors should be able to address most problems during the rebuttal time.

====

After the rebuttal, most of my concerns have been addressed, and I trust the authors to add the extra changes they have promised to improve the paper.

**Q9 Complying With Reviewing Instructions:**

1: Yes.

---

### Official Review · Reviewer_UcK1 · 2022-04-12

**Q2(1) Originality/Novelty:** 2
**Q2(2) Significance/Impact:** 2
**Q2(3) Correctness/Technical Quality:** 3
**Q2(6) Clarity Of Writing:** 3
**Q6 Overall Score:** 6
**Q8 Confidence In Your Score:** 3

**Q1 Summary And Contributions:**

This paper proposed a parametrized model with invariant weights which discovers the invariance in the data by minimizing a lower bound of marginal likelihood. The main contribution of this paper was such a lower bound of marginal likelihood that allows the optimization for the weights of a neural network. And this work applied transformations to the weights rather than the input space.


**Q2 Assessment Of The Paper:**

More detailed information regarding each of these aspects is given below:

**Q2(4) Quality Of Experiments (Optional):**

3: Good: The experimental evaluation is adequate, and the results convincingly support the main claims.

**Q2(5) Reproducibility:**

2: Fair: Key resources (e.g., proofs, code, data) are unavailable but key details (e.g., proof sketches, experimental setup) are sufficiently well-described for an expert to confidently reproduce the main results.

**Q3 Main Strengths:**

1. The idea is appealing and novel that the author applied transformation functions to the weights first instead of the input data, which leads to an invariant MLP. Furthermore, it incorporated common invariant neural structures, such as CNN, into the proposed MLP structure.
2. The derived ELBO allows for simple stochastic gradient descent for the optimization of the marginal likelihood. The theory is elegant and well compatible with algorithm.


**Q4 Main Weakness:**

1. There is abundant related literature on both learning invariant functions and learning invariance from data. This paper’s contribution was only to learn invariance for weights instead of augmented data. The author did not clarify the strength of this approach. Thus, I have doubt over the significance of this contribution.
2. The experiment part lacked baseline methods other than MLP. Since the author mentioned numerous related methods for learning invariance, the according performances should be shown.

**Q5 Detailed Comments To The Authors:**

1. Referring to Q4(1), the author may elaborate with theory, illustration or experiments over the significance of applying transformation to weights rather than input data.
2. Referring to Q3(1), the author may illustrate the generalizability of the proposed method with more common invariant structures in neural network.

**Q7 Justification For Your Score:**

I’m impressed by the novel design of learning invariant parametrized models which automatically mines invariant structures from data. (Q3.1) However, more theory or experiments could be supplemented for the necessity or motivation for this design.


**Q9 Complying With Reviewing Instructions:**

1: Yes.

---

### Official Review · Reviewer_t35N · 2022-04-16

**Q2(1) Originality/Novelty:** 3
**Q2(2) Significance/Impact:** 3
**Q2(3) Correctness/Technical Quality:** 3
**Q2(6) Clarity Of Writing:** 3
**Q6 Overall Score:** 7
**Q8 Confidence In Your Score:** 2

**Q1 Summary And Contributions:**

The paper proposes a marginal likelihood based learning procedure to learn neural networks that are invariant to certain kinds of transformations. The paper is cleanly written, hits all the important points about why each part of the method is required, and shows good results

**Q2 Assessment Of The Paper:**

More detailed information regarding each of these aspects is given below:

**Q2(4) Quality Of Experiments (Optional):**

3: Good: The experimental evaluation is adequate, and the results convincingly support the main claims.

**Q2(5) Reproducibility:**

3: Good: Key resources (e.g., proofs, code, data) are available and key details (e.g., proofs, experimental setup) are sufficiently well-described for competent researchers to confidently reproduce the main results.

**Q3 Main Strengths:**

The paper proposes a new, useful method that seems to work well.

**Q4 Main Weakness:**

It seems a bit heavy to learn a posterior over neural network weights. Can the authors comment on how much more training time was required for their method over the point-estimate?

Further, could the authors point me to why fixed collection of eta (like equally spaced between [0,MAX]) would not help? How much worse was this compared to what the authors propose? Further, was there a baseline with point-estimate theta and fixed collection of eta?

I'm not sure I buy the hypothesis that the authors have about why MLE alone is not enough. Marginal liklihood learning simpler models needs some more evidence. Further, I am not sure the invariance (in MNIST for examples) will lead to higher NLL on the actual data (table 1 for example), so I don't understand why point estimates don't work.

**Q5 Detailed Comments To The Authors:**

See weaknesses.

**Q7 Justification For Your Score:**

Seems like a useful method. a bit heavy but could be a useful stepping stone for build invariant methods using learned invariances. Some confusion about the experiments.

**Q9 Complying With Reviewing Instructions:**

1: Yes.

---

### Decision · Program_Chairs · 2022-05-15

**Decision:**

Accept (Oral)

**Comment:**

Meta Review: The reviewers agree that the proposed idea is appealing with the support of theoretical justification and empirical evidence, the authors provide a fairly strong rebuttal to address most of the concerns raised by the reviewers. I hope the authors will find the review comments helpful for further improving the paper.